# The extraperitoneal French AmbUlatory cesarean section technique leads to improved pain scores and a faster maternal autonomy compared with the intraperitoneal Misgav Ladach technique: A prospective randomized controlled trial

Kaouther Dimassi[1,2]*, Ahmed Halouani[1,2], Amine Kammoun[1], Olivier Ami[3], Benedicte Simon[4], Luka Velemir[5], Denis Fauck[6], Amel Triki[1,2]

1 Obstetrics and Gynecology Department, Mongi Slim University Hospital, La Marsa, Tunisia, 2 Faculty of Medicine, University Tunis El Manar, Tunis, Tunisia, 3 Ramsay Healthcare France, La Muette, Paris, France, 4 Ramsay Healthcare France, Les Franciscaines, Versailles, France, 5 Lenval Foundation Polyclinique Santa Maria, Nice, France, 6 Ramsay Healthcare France, Saint Lambert, La Garenne-Colombes, France

* kaouther.dimassi@fmt.utm.tn

## Abstract

### Objective

To determine whether the French AmbUlatory Cesarean Section (FAUCS) technique reduces postoperative pain and promotes maternal autonomy compared with the Misgav Ladach cesarean section (MLCS) technique in elective conditions.

### Study design

One hundred pregnant women were randomly, but in a non-blinded manner, assigned to undergo FAUCS or MLCS. The primary outcome was a postoperative mean pain score (PMPS), and secondary outcomes were a combined pain/medication score, time to regain autonomy, surgical duration, calculated blood loss, surgical complications, and neonatal outcome.

### Results

Women in the FAUCS group experienced less pain than those in the MLCS group (PMPS = 1.87 [1.04–2.41] vs. 2.93 [2.46–3.75], respectively; p < 0.001). Six hours after surgery, the combined pain/medication score for FAUCS patients was 33% lower than that for MLCS patients (p < 0.001). FAUCS patients more rapidly regained autonomy, with 94% reaching autonomy within 12 h vs. 4% of MLCS patients (p < 0.001). There were no differences in maternal surgical or neonatal complications between groups.

### Conclusions

Our results indicate that FAUCS can reduce postoperative pain and accelerate recovery, suggesting that this technique might be superior to MLCS and should be more widely used.

**Data Availability Statement:** All relevant data are within the manuscript and its Supporting Information files.

**Funding:** The author(s) received no specific funding for this work.

**Competing interests:** The authors have read the journal's policy, and the authors of this study have the following competing interests to share: OA, BS, and DF work for Ramsay Health Care. However, none of the authors receive a salary for their work, nor did they receive any funding for this research. This does not alter our adherence to PLOS ONE policies on sharing data and materials. There are no patents, products in development or marketed products associated with this research to declare.

One potentially key difference between FAUCS and MLCS is that MLCS includes 100 mcg spinal morphine anesthesia in addition to the same anesthesia used by FAUCS. Any interpretation of apparent differences must take the presence/absence of morphine into account.

## Introduction

Cesarean section (CS) is the most common major obstetric surgery and the oldest operation in the field of abdominal surgery. Until the 17th century, CS was an exclusively lethal operation for the mother, performed to save or separate the newborn from a dead or dying mother [1,2]. Fortunately, improvements in obstetric surgical techniques based on modern scientific concepts were achieved at the end of the 20th century, leading to safer, simpler, and less traumatic approaches to CS [2]. In the 1990s, Stark introduced the Misgav Ladach cesarean section (MLCS) technique [3], also known as the Joel-Cohen technique, which became a widely-used standard for CS delivery in Tunisia and several European countries.

Recently, little progress has been made in the field of CS techniques. Twenty years after their invention, intraperitoneal CS techniques such as MLCS remain widely used [4]. Today, approximately 1 in 5 births worldwide is performed by CS due to various factors influencing medical decision-making [5] and their intuitive protective effect in cases of cephalopelvic dystocia [6]. However, CS techniques remain pathogenic and traumatic compared with vaginal delivery. Women who undergo CS frequently experience severe pain, feel disabled, and are unable to take care of themselves or their newborn in the first days after surgery, which affects maternal-infant bonding and the ability to lactate [7,8]. However, with advances in woman-centered perinatal care, a return to extraperitoneal approaches is regaining interest [9,10], because they are associated with less need for intravenous painkillers, shorter hospital stays, and earlier return to home, where breastfeeding can occur in a more comfortable environment.

In a retrospective analysis of over 3,000 cases, an innovative approach to CS known as the French AmbUlatory cesarean section (FAUCS) was found to enhance women's recovery [9]. FAUCS differs from MLCS in several major ways. MLCS involves a horizontal opening of the aponeurosis and linea alba splitting on the median line, tearing of the peritoneum with fingers, and an intraperitoneal approach allowing air, blood, and amniotic fluid into the peritoneal cavity. Uterine suture is performed with a single-layered linear suture along the entire hysterotomy. In the FAUCS technique, however, only the anterior sheath of the aponeurosis is opened vertically on the left side, which is called a paramedian incision. Given no posterior sheath exists under Douglas' line, the linea alba is respected. The approach is extraperitoneal on the left side of the bladder, and the hysterotomy is closed using a double-layered purse-string suture.

We introduced the FAUCS technique to our maternity department in January 2018 and demonstrated its safety in a previous study [4]. In the present study, we determined whether FAUCS reduces postoperative pain compared with MLCS without increasing intra- or postoperative complications in elective conditions.

## Materials and methods

### Ethics statement

This study was specifically approved by Mongi Slim University Hospital, La Marsa, Tunisia, local hospital ethics committee (approval number 05/2018) on March 5th, 2018 and written consent was obtained. Enrolment of the first patient started on August 1st 2018.

Because all protocols were originally written in French, this study was registered on clinical-trials.org (NCT03741907) on November 15[th] 2018, after enrolment of participants started, due to administrative and english translation process delay, but registration at "clinicaltrials.gov" is not mandated by Tunisian law, which covered all legal aspects of this study. The full protocol is available on the clinicaltrials.gov site.

Data collection was conducted in compliance with Tunisian laws regarding personal data protection. The authors confirm that all ongoing and related trials for this procedure are registered.

## Study design and participant selection

We performed an unblinded randomized clinical trial comparing MLCS versus FAUCS at Mongi Slim University Hospital, La Marsa, Tunisia, between August 2018 and March 2019. Women were included if they had a singleton term pregnancy delivered through a planned indicated CS. Women were excluded if their pregnancies involved known fetal, placental, or uterine anomalies.

All women who met the inclusion criteria, including being age 18–48 and gestational age 37 weeks, were invited to participate in the study during their final prenatal visit. Recruitment started on August 1[st] of 2018 and follow-up of the last enrolled patient ended on March 31[st] of 2019. Those providing written informed consent were consecutively included in a preliminary patient list managed by an investigator who was not involved in patient care. Before randomization, investigators excluded women who were initially recruited but had to undergo emergency surgery before the originally scheduled date (e.g., in cases of acute fetal compromise) or were operated on by a different surgeon than those assigned to the study. Women included in the study were provided a study number on their delivery day in chronological order. Random assignment to FAUCS or MLCS groups was performed by an investigator who was not involved in patient care using Kendall and Smith's Tables of Random Sampling Numbers [11]. Subjects were assigned into 2 blocks of 50 subjects for randomization.

## Surgical procedures and after-care

Participants, residents involved in patient care, and caregivers were blinded to the CS technique before the operation and were informed at discharge. Anesthetists and surgeons were informed of the CS technique on the due date in the operating room. No post-op caregivers and residents were present during the surgery.

The MLCS spinal anesthesia protocol included 7–10 mcg bupivacaine (depending on patient height), 100 mcg morphine, and 10 mcg sufentanil. The FAUCS spinal anesthesia protocol included 7–10 mg bupivacaine (depending on patient height), 10 mcg sufentanil, and no morphine.

All surgeries were performed by two senior surgeons, who performed the MLCS technique as described in 1999 [3] or the FAUCS technique as described in 2017 [9]. The main differences between FAUCS and MLCS concerned fascia incision, approach of the lower uterine segment, and uterine closure [4]. In the MLCS technique, tissues and fascia were spread apart ~2–3 centimeters at the midline, and the incision was further broadened with two fingers. The vertical rectus muscles were separated, and the peritoneum was opened transversely with fingers. The uterus was closed with a single-layer suture. In the FAUCS technique, only the anterior sheath of the aponeurosis was opened vertically on the left side, and the linea alba was respected. The approach was extraperitoneal on the left side of the bladder. The uterus was closed using a purse-string suture [4,9]. To ensure that the patients who had undergone

FAUCS or MLCS were visually indistinguishable post-operation, a subcuticular absorbable Vicryl suture was employed for skin closure in both techniques.

Hematocrit levels were assessed before and after surgery. Analgesics were administered by nursing staff upon patient request via a visual analog scale (VAS) [12]. VAS is the most used scale for assessing level of post-operative patient pain. VAS assessments were made every 6 h: H0, H6, H12, H18, and H24. A standardized analgesic scheme was used: first line was intra-rectal 100 mg ketoprofen every 6 h; second line was intravenous 1 g paracetamol every 6 h, and third line was oral 50 mg tramadol every 6 h.

Women were encouraged to make an attempt to stand every hour after surgery. Normal oral food intake was initiated as soon as gas passage occurred and when the patient felt hungry. Women were evaluated 24 h after surgery. If there were no complications and if the patient felt autonomous and pain-free, she was discharged 24 h after surgery. In other cases, patients were discharged at least 48 h after surgery after a similar evaluation of maternal autonomy.

## Study outcomes

The primary outcome measure was the postoperative mean pain score (PMPS), calculated as a mean of the five VAS scores performed at postoperative time points every 6 h over the course of 24 hours: H0 which designated the end of surgery time, H6, H12, H18 and H24.

Secondary outcome measures were as follows: (1) postoperative pain and medication score (PAMS), calculated as a combined pain/medication score, every 6 h as $\left(VAS + \frac{ketoprofen\ dose}{100}\right) \div 2$; (2) total surgical duration in min; (3) calculated blood loss (CBL), derived as $BV_M \times \%BV_\Delta$, where $BV_M$ is maternal blood volume calculated using Nadler's formula [13] and $\%BV_\Delta$ is percent change (i.e., loss) of blood calculated using Brecher's formula [13]; (4) maternal autonomy, calculated as a last of the postoperative times to spontaneous urination, standing, or first meal in hours—whichever came last; and (5) time to discharge in days. Newborn outcomes measures were: (1) appearance, pulse, grimace, activity, and respiration (APGAR) score at 1, 3, 5, and 10 min; (2) acid-base balance/eucapnic pH [14,15]; and (3) rate of hospitalization.

## Statistical analysis

Sample size calculation was based on the PMPS. The effect of FAUCS was considered important if the difference in PMPS between groups was $\geq$ 30%, using a t-test. To achieve 80% power with an $\alpha = 0.05$, it was anticipated that 45 patients in each group were required.

Data were analyzed by generalized linear models for which Nagelkerke's pseudo-$R^2$ was calculated. Combined pain/medication score was analyzed as a repeated measure with generalized mixed-level linear models testing "technique + time" and "technique + time + (technique × time)", with individual patient as a random effect. Maternal autonomy was analyzed by a Cox proportional hazard model with patient as a gamma-distributed frailty term. Surgical duration, CBL, days to discharge, and eucapnic pH were analyzed as continuous variables using generalized linear models. Incidence of neonate hospitalization was analyzed using a logistic generalized linear model. The APGAR score was analyzed using cumulative link mixed models.

## Results

Of the 487 women who underwent a CS during the study period, 169 were planned and were assessed for eligibility. One hundred women were randomized into the FAUCS or MLCS group. After randomization, all patients completed the study and were included in the analysis

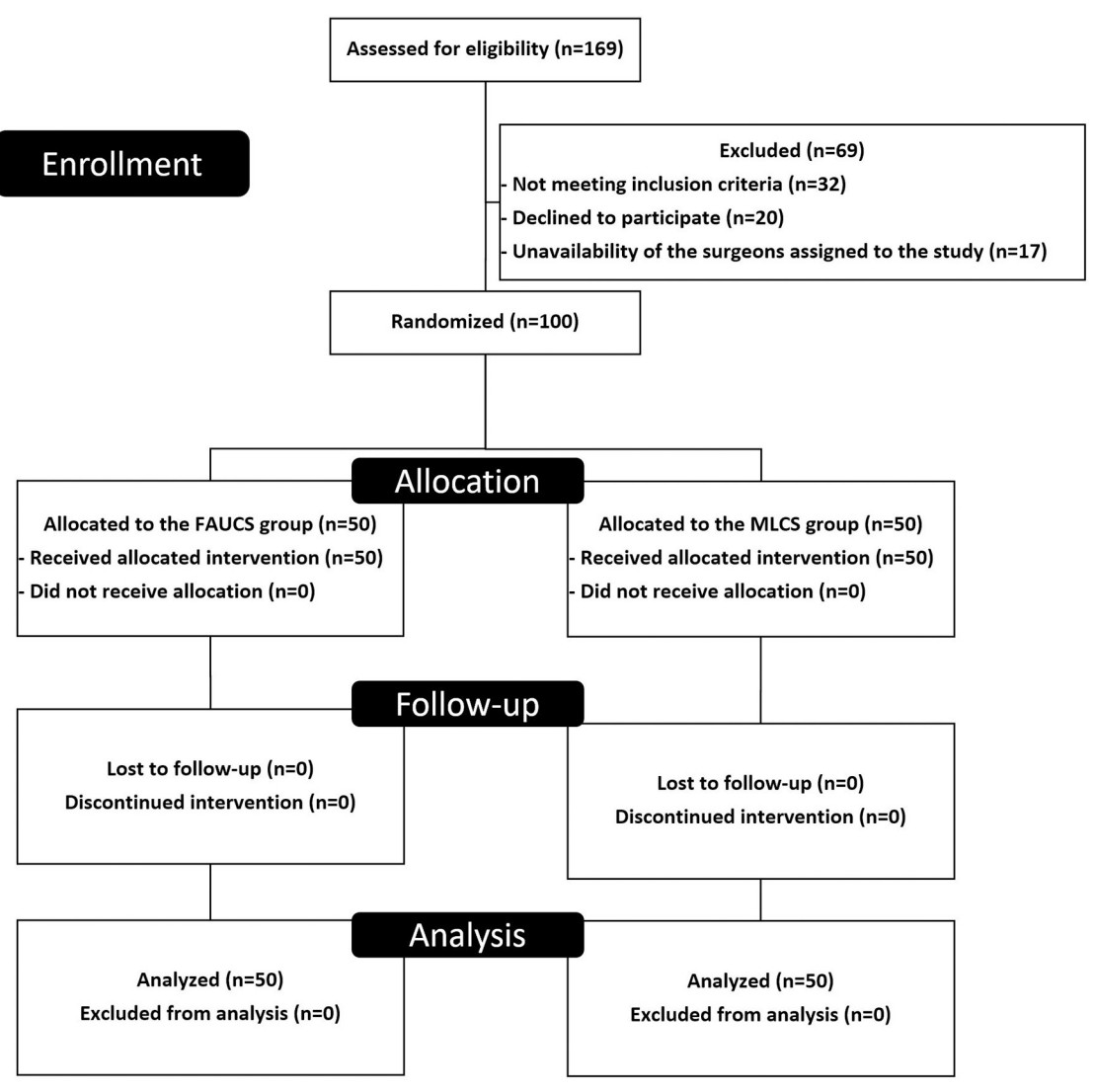

**Fig 1. CONSORT 2010 flow diagram.**

(Fig 1). There were no significant differences between groups in women's epidemiological or obstetric characteristics (Table 1).

FAUCS could be performed for all patients who previously underwent MLCS; in these cases, tissues were mostly non-cicatricial for the left paravesical extraperitoneal approach. No patients had previously undergone FAUCS prior to this study. No cases needed to be excluded intraoperatively.

Overall, postoperative pain was significantly higher in MLCS patients (PMPS [first–third quartile], 2.93 [2.46–3.75]) than in FAUCS patients (1.87 [1.04–2.41]; $p < 0.001$). However, the effects of technique and time and their interaction were all significant ($p < 0.001$, $p < 0.001$, and $p = 0.001$, respectively). Specifically, 6 h after surgery, the PAMS was 33% lower for FAUCS patients than for MLCS patients (Fig 2). Although PAMS for both groups converged over time, scores remained lower for FAUCS patients than for MLCS patients at each time point (Table 2). Also, FAUCS allowed a faster return to autonomy, with 94% of FAUCS patients attaining autonomy by 12 h compared with 4% of MLCS patients.

**Table 1. Patient characteristics.**

| Characteristic | MLCS group[a] | FAUCS group[a] | Test statistic[b] | p-value[c] | R²[d] | Raw p-value |
|---|---|---|---|---|---|---|
| Weight (kg) | 82.95 ±1.99 | 79.52 ± 1.69 | 1.741 (1, 98) | 1.000 | 0.018 | 0.190 |
| Height (m) | 1.61 ± 0.01 | 1.62 ± 0.01 | 0.065 (1, 98) | 1.000 | 0.000 | 0.799 |
| BMI | 31.79 ± 0.61 | 30.42 ± 0.55 | 2.758 (1, 98) | 0.900 | 0.027 | 0.100 |
| Age (years) | 33.86 ± 0.75 | 32.80 ± 0.77 | 0.975 (1, 98) | 1.000 | 0.010 | 0.326 |
| Gestation (weeks) | 39.08 ± 0.10 | 39.14 ± 0.10 | 0.185 (1, 98) | 1.000 | 0.002 | 0.668 |
| Pre-operative hematocrit | 33.82 ± 0.49 | 33.68 ± 0.45 | 0.040 (1, 98) | 1.000 | 0.000 | 0.841 |
| Gravidity | 2.5 ± 0.5 | 2.5 ± -2 | 0.231 (1) | 1.000 | 0.018 | 0.631 |
| Parity | 2 ± 1/0 | 2 ± -1/2 | 1.730 (1) | 1.000 | 0.002 | 0.188 |
| Prior CS | 1 ± 1/0 | 1 ± -0.25/0 | 0.628 (2) | 1.000 | 0.007 | 0.730 |

[a]Mean ± Standard error of mean (SEM) for weight, height, body mass index (BMI), age, gestation, and hematocrit; Median ± 75th/25th percentiles for gravidity, parity, and prior CS.

[b]F (df$_{numerator}$, df$_{denominator}$) for weight, height, BMI, age, gestation, and hematocrit; likelihood ratio $\chi^2$ (df) for gravidity, parity, and prior CS.

[c]Holm-adjusted for family-wise error rate, nine simultaneous tests.

[d]Nagelkerke's pseudo $R^2$.

All patients, regardless of group, attained autonomy by 24 h post-delivery (Fig 3). However, FAUCS patients recovered more rapidly than MLCS patients (p < 0.001), with 94% of FAUCS patients attaining autonomy by 12 h compared with 4% of MLCS patients (Table 3). MLCS patients were discharged in 1.88 ± 0.05 (mean ± SEM) days versus 1.18 ± 0.07 days for FAUCS patients (p < 0.001).

Total surgical duration (38.38 ± 2.24 vs. 43.31 ± 7.34 min, p = 0.414) and CBL (520 ± 58 vs. 536 ± 50 ml, p = 0.724) were similar in MLCS and FAUCS patients, respectively. The hypothesis tested was that the durations would differ. Instrument assistance using forceps or spatulas

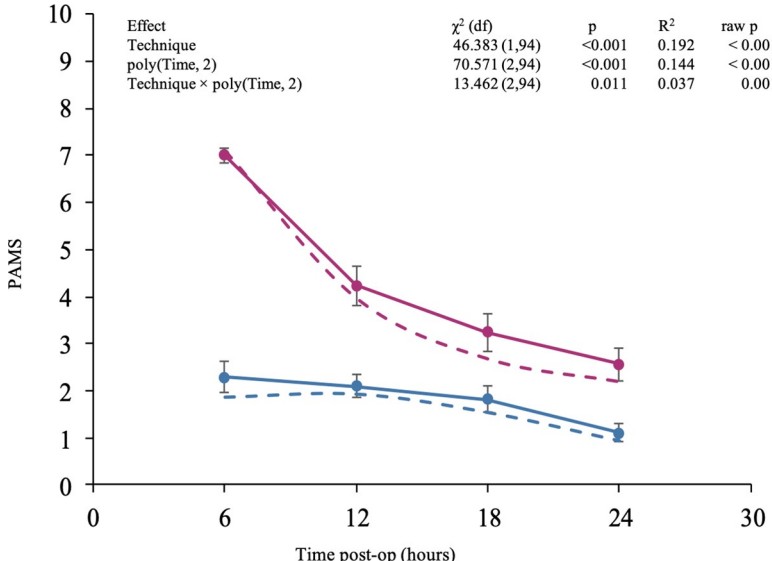

**Fig 2. General linear model analysis of PAMS score at selected time points post-CS.** Hypothesis tested was whether overall PAMS differed between FAUCS and MLCS (Technique variable), PAMS would decrease over time (Time variable), and the rates of decrease (interaction) would differ between FAUCS and MLCS. Data are shown as mean ± SEM. Dashed lines represent model predictions for fixed effects (technique, time, and technique × time). Effect of Technique χ2 (df) = 46,38 (1,94); p < 0.001; R2 0.192; raw p < 0.001. Effect of Time χ2 (df) = 70,57 (2,94); p < 0.001; R2 .144; raw p < 0.001. Effect of Technique x Time χ2 (df) = 13,46 (2,94); p = 0.011; R2 0.037; raw p = 0.001.

**Table 2. PAMS (mean ± SEM) at various time points post-CS.**

| Time (h) | MLCS group | FAUCS group | Mean difference |
|---|---|---|---|
| 6 | 7.00 ± 0.14 | 2.28 ± 0.33 | 4.72 ± 0.46 |
| 12 | 4.23 ± 0.42 | 2.10 ± 0.23 | 2.13 ± 0.32 |
| 18 | 3.24 ± 0.40 | 1.82 ± 0.27 | 1.42 ± 0.39 |
| 24 | 2.56 ± 0.36 | 1.10 ± 0.18 | 1.46 ± 0.26 |

was necessary in 84% of FAUCS procedures (p < 0.001). The hypothesis tested was that FAUCS would result in increased instrument assistance, as FAUCS incisions are smaller and instrument assistance usually described as part of the technique.

Newborn outcomes did not differ by surgical procedure. Visual inspection after a cumulative link mixed model analysis of APGAR scores appeared to show a slightly beneficial effect of FAUCS, which was associated with an APGAR score of 10 at 1 min. However, the effect size for this difference was small (Cramer's V = 0.123 at 1 min) and grew smaller over time (Cramer's V = 0.096, 0.060, and 0.000 at 3, 5, and 10 min, respectively). Eucapnic pH and frequency of hospitalization were similar in both groups (p = 0.714 and p = 1.000, respectively).

## Discussion

We found that the FAUCS technique described by Ami et al. [9] reduces postoperative pain without increasing intra- or postoperative complications compared with MLCS in elective

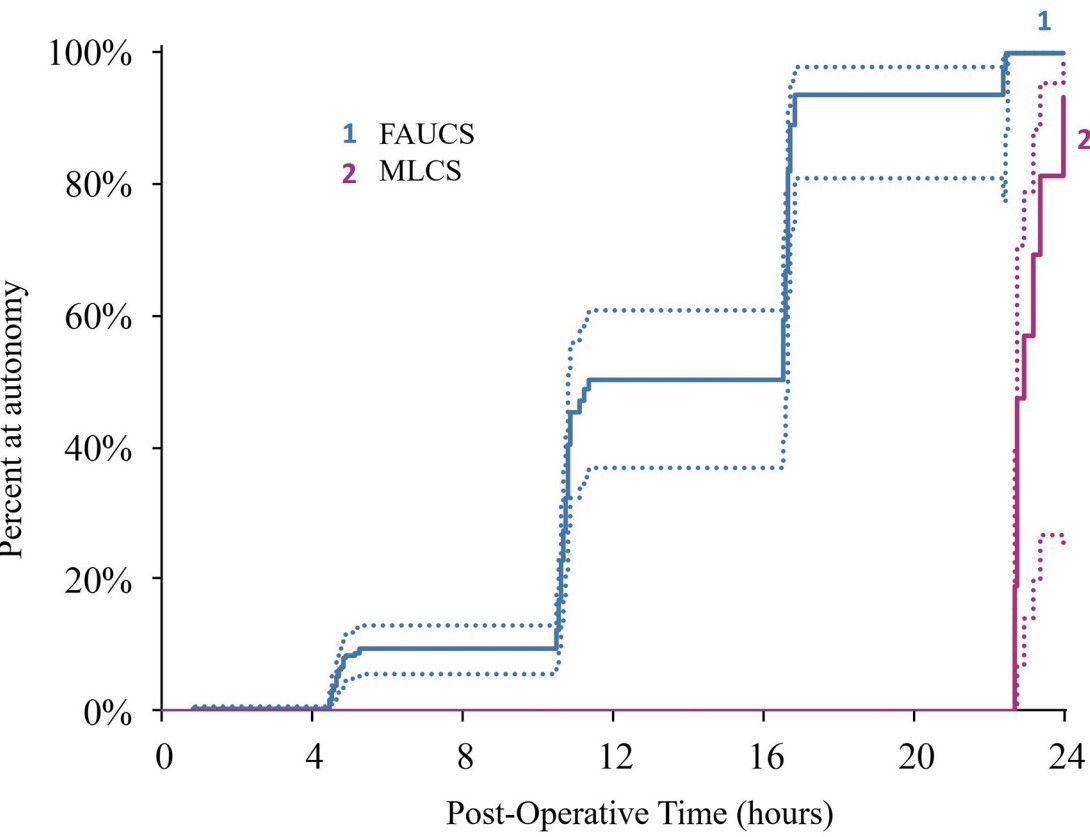

**Fig 3. Cox proportional hazards model analysis of percentage of patients exhibiting autonomy at selected time points post-CS.** Hypothesis tested was that FAUCS would result in more rapid gain of the "autonomy" measure. Effect of Technique $\chi 2$ (df) = 625 (1); p < 0.001; R2 0.174; raw p < 0.001. Effect of frailty (patient) $\chi 2$ (df) = 490 (1); p < 0.001; R2 .758; raw p < 0.001.

**Table 3. Percentage of patients exhibiting autonomy at various time points.**

| Time (h) | MLCS group | FAUCS group |
|---|---|---|
| **2** | 0% | 2% |
| **6** | 0% | 44% |
| **12** | 4% | 94% |
| **18** | 28% | 98% |
| **24** | 100% | 100% |

conditions. FAUCS also enhanced women's autonomy after surgery, thus shortening their hospital stay.

The sample size assumptions were realized in this clinical trial, with an initial hypothesis of 30% reduction of PMPS, and an effective 36,17% of PMPS reduction observed.

For both pain and autonomy measures, the advantageous effect of FAUCS over MLCS was higher at earlier time points after surgery. From both patient and medical management standpoints, these are critical outcomes.

Given that the rate of planned CS is increasing in many countries, obstetric teams are working to enhance women's recovery and reduce their length of hospital stay [10]. The most common steps for enhancing recovery after surgery are early oral intake, early mobilization, early removal of catheters, patient advice and information, and regular postoperative analgesia. A previous study has shown that an enhanced recovery program for women undergoing planned CS increases the proportion of women discharged on day 1 from 1.6% to 25.2% [16]. In most settings, hospitalization after a CS delivery is 3–4 days, which could make it difficult to envision women being discharged 24 h after surgery. However, this was achieved in the previous study using adequate analgesia and follow-up at home without making any improvements to the surgical technique. In the present study, improvements in women's outcomes after CS were achieved by the surgical technique itself. Specifically, 86% of women who underwent FAUCS were discharged on day 1. It could be speculated that immediate mobilization after FAUCS could help prevent venous thromboembolism and promote early mother-baby bonding. Furthermore, the absence of peritoneal cavity opening allows earlier oral food intake. All of these beneficial short-term outcomes could also improve long-term outcomes, such as preservation of future fertility, by preventing intraperitoneal adhesions.

The FAUCS technique involves multiple innovations that might hinder its diffusion. However, each of these innovations has the potential to bring about benefits. First, left paramedian incision could benefit from the natural mechanical behavior of the abdominal wall. The stiffest structures, specifically the aponeuroses and linea alba, are those that perform the most work in the abdomen. Thus, the linea alba is the most important unit contributing to the mechanical stability of the abdominal wall [17]. Greater compliance of the linea alba, strain on the intact abdominal wall, and stiffness of the rectus sheath and umbilical fascia all exist when tissues are loaded in the longitudinal direction compared with the transverse direction [18]. Additionally, greater stress is placed on the linea alba when it is loaded in the transverse direction compared with that in the longitudinal direction [18]. The lateral paramedian incision is slightly more time-consuming to perform but results in a significantly lower incidence of incisional hernia [19]. When a vertical abdominal incision is being considered, the lateral paramedian should be the incision of choice [19]. In our study, a paramedian incision contributed to women's quick recovery after CS. Further studies are needed to confirm this impact on mid- and long-term maternal autonomy.

Extraperitoneal CS is a method of surgically delivering a baby through an incision in the lower uterine segment without entering the peritoneal cavity, given keeping the peritoneal

cavity intact reduces the risk of adhesions, postoperative ileus, and future infertility related to surgery [20]. In the FAUCS technique, the uterus is approached through the paravesical space, which can allow the earlier return of bowel function, as evidenced by the higher autonomy scores among FAUCS patients than among MLCS patients. Similar results are reported in randomized trials, which show that bowel function returns at an average of 8 ± 4 h in extraperitoneal CS compared with 13 ± 4 h in transperitoneal CS [20]. Extraperitoneal CS also reduces the use of intravenous fluids and analgesics without increasing surgical complications [4,9,20]. In our study, we observed no complications in either group. Since the implementation of FAUCS in our study unit [4], we have experienced three bladder injuries out of 200 cases. In a retrospective study introducing the FAUCS technique [9], 11 out of 3,441 cases (0.3%) involved bladder injury. However, bladder injuries during CS are not specific to the extraperitoneal approach and have an overall incidence of 0.44% [21].

During FAUCS, the combination of a horizontal skin incision, paramedian vertical aponeurotic opening, and extraperitoneal approach leads to a relatively small extraction field [9]. Consequently, instruments are often used to facilitate fetal extraction [4,9], as was done for 84% of our FAUCS patients. However, this appeared to have no impact on neonatal outcomes, given fetal blood eucapnic pH and the frequency of neonatal hospitalization did not differ between FAUCS and MLCS.

We recommend some tips for reducing extraction time while performing FAUCS, such as: (1) opening the anterior rectus sheath with sufficient width up and down; (2) reclining the left rectus abdominis muscle correctly to the left; and (3) involving the mother in the process by having her control her breathing [4,22] during the use of small guiding instruments (e.g., Wrigley forceps or Teissier spatulas) [23] which allows to perform smaller incisions. This instrument assistance is usually very easily and gently performed, while there is no bony obstacle to the exit of the newborn, but only soft tissues around.

Uterine incision closure is the most important factor contributing to good healing and preventing future CS-related complications. With continuous single-layer uterine closure, uterine incisional defects occur in 20%-60% of cases [24]. It is reasonable to believe that uterine scarring defects reflect poor or incomplete healing of part of the hysterotomy. The mechanism of this defective healing could be the mechanical tension of the lower uterine segment, which might impair blood perfusion and oxygenation of healing tissues. To reduce mechanical tension in the lower uterine segment, purse-string suturing has been used to remove myomas during CS [25]. Purse-string suture of the uterine incision provides good control of bleeding and decreases the length of the uterine wound while increasing its thickness [9]. Unlike the Turan technique described in 2015 [24], FAUCS uterine closure uses a double-layer purse-string closure. A previous study has shown that with the purse-string closure technique, uterine incision length is shorter (3.7 cm vs. 8.5 cm) and uterine scar defect frequency is lower (23.5% [12/51] vs. 60% [39/65]) than that for the traditional double-layer uterine closure technique [9]. The high frequency of uterine incision defects in the previous study might have been because the incision site was examined only 6 weeks after surgery. A second study with the same study population is currently in progress to evaluate purse-string closure 6 months after surgery (NCT03930134).

In the present study, total surgical duration was similar between groups (38.38 ± 2.24 min for MLCS vs. 43.31 ± 7.34 min for FAUCS; p = 0.414), consistent with our observation of no difference in CBL. However, during the FAUCS learning curve [4], surgical duration is longer for the FAUCS technique than for the MLCS technique (50 [40–60] vs. 35 [30–40] min, respectively, p < 0.001) [4].

This study has potentially significant limitations. In particular, the anesthesia protocols for the two techniques differed. Thus, the absence of the adverse effects of morphine in the

FAUCS group (e.g., nausea, delayed transit, urinary sphincter retention) could have led to improved outcomes. Morphine in the epidural has long been considered necessary in our anesthesiology department as part of the early post-surgery rehabilitation protocol, given it effectively reduces pain on the first day and facilitates early mobilization. However, we excluded it from the FAUCS protocol because we have found that this surgical procedure allows early mobility without a high degree of pain, with previous studies also supporting its exclusion [4,9]. While it might be tempting to "fold" potential opiate-specific effects into any comparison against MLCS, we must admit that an unknown and potentially important proportion of differences between MLCS and FAUCS is actually due to morphine. However, such a comparison would require performing MLCS without morphine, which requires particularly ethically sensitive research specific to withholding anesthesia. Such work could be more appropriate to a multi-center study.

In conclusion, our findings indicate that FAUCS reduces pain and enhances recovery compared with MLCS. Therefore, we conclude that FAUCS is a highly desirable method to use in cases of planned CS, is superior to MLCS, and should be implemented on a widespread basis.

## Supporting information

**S1 Checklist. Consort checklist.**
(DOC)

**S1 Dataset.**
(XLSX)

**S1 File. Original trial study protocol.**
(DOCX)

**S2 File. Translation trial study protocol.**
(PDF)

## Acknowledgments

The authors thank Israel Hendler (reviewing and editing the article) and Sivan Navot and Samy Chouial (diffusion of the results) who have made valuable contributions in preparation of this article.

## Author Contributions

**Conceptualization:** Kaouther Dimassi, Denis Fauck.

**Formal analysis:** Kaouther Dimassi, Olivier Ami.

**Investigation:** Ahmed Halouani, Amine Kammoun.

**Methodology:** Kaouther Dimassi.

**Software:** Olivier Ami.

**Supervision:** Kaouther Dimassi, Amel Triki.

**Validation:** Kaouther Dimassi.

**Writing – original draft:** Kaouther Dimassi.

**Writing – review & editing:** Olivier Ami, Benedicte Simon, Luka Velemir, Denis Fauck.

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
