## [Decision Letter · Decision Letter 0]

30 Oct 2020

PONE-D-20-27208

The extraperitoneal French AmbUlatory cesarean section technique leads to superior maternal outcomes compared with the intraperitoneal Misgav Ladach technique: A prospective randomized controlled trial

PLOS ONE

Dear Dr. Kaouther Dimassi,

Thank you for submitting your manuscript to PLOS ONE. After careful consideration, we feel that it has merit but does not fully meet PLOS ONE’s publication criteria as it currently stands. Therefore, we invite you to submit a revised version of the manuscript that addresses the points raised during the review process.

We look forward to receiving your revised manuscript.

Kind regards,

Georg M. Schmölzer

Academic Editor

PLOS ONE

Journal Requirements:

2. Thank you for submitting your clinical trial to PLOS ONE and for providing the name of the registry and the registration number. The information in the registry entry suggests that your trial was registered after patient recruitment began. PLOS ONE strongly encourages authors to register all trials before recruiting the first participant in a study.

1) your reasons for your delay in registering this study (after enrolment of participants started);

2) confirmation that all related trials are registered by stating: “The authors confirm that all ongoing and related trials for this drug/intervention are registered”.

Please also ensure you report the date at which the ethics committee approved the study as well as the complete date range for patient recruitment and follow-up in the Methods section of your manuscript.

3.  Thank you for including your ethics statement:  "ethics committee mongi slim university hospital , la marsa tunisia

approval number 05/2018

written consent"

Please amend your current ethics statement to confirm that your named institutional review board or ethics committee specifically approved this study.

"NO authors have competing interests"

We note that one or more of the authors are employed by a commercial company: Ramsay Healthcare France.

4.1. Please provide an amended Funding Statement declaring this commercial affiliation, as well as a statement regarding the Role of Funders in your study. If the funding organization did not play a role in the study design, data collection and analysis, decision to publish, or preparation of the manuscript and only provided financial support in the form of authors' salaries and/or research materials, please review your statements relating to the author contributions, and ensure you have specifically and accurately indicated the role(s) that these authors had in your study. You can update author roles in the Author Contributions section of the online submission form.

4.2. Please also provide an updated Competing Interests Statement declaring this commercial affiliation along with any other relevant declarations relating to employment, consultancy, patents, products in development, or marketed products, etc.  

Reviewers' comments:

Reviewer's Responses to Questions

**Comments to the Author**

1. Is the manuscript technically sound, and do the data support the conclusions?

Reviewer #1: No

Reviewer #2: Yes

2. Has the statistical analysis been performed appropriately and rigorously? 

Reviewer #1: Yes

Reviewer #2: Yes

3. Have the authors made all data underlying the findings in their manuscript fully available?

Reviewer #1: No

Reviewer #2: Yes

4. Is the manuscript presented in an intelligible fashion and written in standard English?

Reviewer #1: Yes

Reviewer #2: Yes

5. Review Comments to the Author

Reviewer #1: The authors present the results of a single center randomized trial of two different cesarean delivery techniques to determine if the French AmbUlatory cesarean (FAUCS) technique resulted in significantly lower postoperative mean pain scores than a more standard technique used at their hospital, the Misgav Ladach (MLCS) approach. Other outcomes included a combined pain and medication score, total surgical duration, calculated blood loss, maternal autonomy which included measures of time to urination, standing and eating meals, and length of stay. One of two surgeons performed all procedures. Unfortunately, the anesthetic techniques differed between the two groups. The authors conclude that the FAUCS technique “can reduce postoperative pain and accelerate recovery” when compared to the MLCS approach.

Questions and comments for the authors.

1. Inclusion/Exclusion criteria. The general summary statement on the Clinicaltrials.gov site states that women at 36 weeks’ gestation and greater were eligible. On the Clinicaltrials.gov site there is also a link to PDF titled, “Study Protocol and Statistical Analysis Plan”. In this document, the inclusion criteria section states that women 37 weeks’ and greater were eligible. Next, the manuscript does not specifically state the gestational age at which women were eligible. Please add to the manuscript the gestational age inclusion criteria and clarify the discrepancy between the summary on the Clinicaltrials.gov site and that in above-referenced study protocol PDF, which is a link on the CT.gov site. The manuscript also does not list the age that women had to be in order to be eligible (CT.gov site states 18-48 years).

2. Randomization scheme. The Materials and Methods section of the manuscript (page 5) states that “The two comparison groups were generated using simple randomization, with an equal allocation ratio, by referring to Kendall and Smith’s Table of Random Sampling Numbers [11]”. I went to the referenced web site and the text book was not available on the listed link. Please provide more explanation on the randomization scheme. What were the block sizes? Were blocks used or was the randomization just alternating group assignments. This is important because if blocks were not used and the randomization was just simple alternating numbers, the providers could guess which technique was next and it is possible then that the subject could have been informed of the allocation group.

3. Blinding. The second paragraph of page 5 states that you performed an “unblinded randomized clinical trial” but then on page 6, in the first paragraph you state “participants, residents involved in patient care, and all investigators were blinded”. These two statements seem contradictory. I appreciate that the surgical team cannot be blinded to the cesarean but then you state that “all investigators were blinded”. If the surgeons are co-authors on the manuscript, then I would consider them as investigators. Next, were residents who performed the post-operative care not present or scrubbed for the surgery? Please clarify the statements about blinding. I worry greatly that patients could have discovered which group assignment they were in and if they then also knew that FAUCS was the intervention being tested, that they would then want to report improved pain scores.

4. Anesthetic plan. Each group received a different anesthetic regimen. You acknowledge this limitation (page 16), but I believe this to be a significant weakness in your study. The anesthetic regimens may be driving your outcomes rather than the surgical procedure. By having different anesthetic regimens between the two groups, you lose the benefit of performing a randomized trial that attempts to test the difference between two surgical techniques. With this design, it is impossible to know if the different surgical techniques are driving the findings, or the different anesthetic techniques. This limitation needs to be highlighted throughout the text, including any concluding statements in the Abstract and Discussion (and title) sections which state that the FACUS resulted in improved postoperative pain.

5. Surgical techniques. Neither the MLCS nor the FAUCS techniques are commonly used in the United States. You provide some references to the techniques and briefly describe them but my preference is that more information be provided on these techniques. Can you work with the Academic Editor on this? Maybe you could include illustrations of the two techniques as Supplemental material. This added information would help obstetricians in the United States better understand the two techniques. Most obstetricians in the US use either a Pfannenstiel (with lateral rectus sheath fascial incisions) or a Joel-Cohen technique.

6. One of the secondary outcomes was listed as “combined pain/medical score”. Which analgesic agent was used for the combined pain/medical score calculation? Earlier in the manuscript, you state that your standard postoperative analgesic plan was intra-rectal ketoprofen and then intravenous paracetamol as second line. Did you just use the ketoprofen medication dose in the calculation used for this secondary outcome? Please clarify.

7. The title states that the FAUCS “leads to superior maternal outcomes”. I would clarify this statement and make it more specific; i.e. improved pain scores. Also see my concerns outlined in #4.

8. You state that your primary outcome is a mean of the five VAS scores performed at postoperative time points, H0, H6, H12, H18 and H24. These five time points were never explicitly defined. I appreciate that they were performed every six hours over the course of 24 hours but when did they start? What is H0? Please clarify.

9. Table 2 and Figure 2 provide mean VAS scores at each of the five time points. The y-axis of the figure and the table legend both use the abbreviation “PMPS”. This is technically not correct. You define the PMPS as the average of the five time points while the figure and table provide the mean of each of the individual VAS scores at each time point. Please clarify and adjust the Figure axes and Table legend to clarify that you are presenting mean VAS scores at each time point and not your defined PMPS.

10. Please provide p-values in Tables 2 and 3 so the reader can determine if the differences in mean VAS scores (Table 2) and autonomy scores (Table 3) are significantly different at each time point. This is particular important as you claim that the FAUCS results in improved pain and autonomy.

11. Table 1 lists Test statistic values, p-values, R2 values and raw p-values. These can all be excluded from the table given that your performed a randomized trial. The Academic Editor can clarify if the journal requires these values.

12. The first full paragraph of the Page 10 (Results section) provides the differences in “postoperative pain” between the two groups. Is this your PMPS score? If so, please clarify.

13. I am confused about the secondary outcome that is labeled “maternal autonomy”. You attempt to define this outcome in the ‘Study Outcomes’ paragraph on page 7 but the definition is still not clear. You state that maternal autonomy was “calculated as a summation of postoperative times to spontaneous urination, standing, or first meal in hours”. Please provide more information on how this variable was calculated. Was it the amount of time from the conclusion of the surgery to any one of those three outcomes? The inclusion of the word “or” suggests that it was the time to any one of the three but you also use the word ‘summation’, which suggests it was the total time to all three. Was the outcome; autonomy = (time from end of surgery to urination) + (time from end of surgery to unassisted standing) + (time from end of surgery to first meal)? Were foley catheters removed at the same duration of time following surgery for all subjects? If not, the variation in when the catheters were removed could affect this outcome rather than the surgical intervention. The delivery of food trays could also be varied based on what time of day the surgery was performed. Since you don’t feed women until they pass flatus, why did you use eating as the outcome variable rather than passing flatus?

14. The last sentence on page 11 states, “Instrument assistance was necessary in 84% of FAUCS procedures (p<0.001)”. What instruments are you referring to? Next, what is the p-value comparing this outcome to? Is it for the use of instruments in the MLCS group?

15. The second paragraph of the Discussion section refers to better overall birth experience and lower financial costs. Neither of these outcomes were studied, thus I would suggest that these claims are removed from the manuscript.

16. I may have missed it, but I did not see that you reported the 'combined pain/medication score' outcome data that was a planned secondary outcome.

17. The third paragraph of the Discussion includes a statement, “We found that FAUCS was so effective at reducing surgical pain that early rehabilitation depended more on preserved maternal autonomy than on painkillers and protocols”. I do not understand this statement. Next, given that the two groups received different anesthetic protocols, you do not know if the differences in postoperative pain scores that were seen were due to the surgical technique or the anesthetic technique.

18. CONSORT. Please state that the full protocol is available on the Clinicaltrials.gov site. See my concerns above about randomization and blindings.

Reviewer #2: This is the report of a randomized clinical trial of Cesarean techniques (I think Cesarean should be capitalized throughout, incidentally). The primary outcome is a VAS composite score. The authors need to reference their score functions, or at least explain why they are appropriate. The randomization procedure is described, statistical analysis is sophisticated with longitudinal and survival models, and limitations of the study are clearly enumerated. The study statistician uses Holm's method to adjust for multiplicities. From a statistical standpoint, the authors are to be congratulated. There are two glaring omissions that would be useful to correct in a revision:

1. Sample size is not really described. 30% of what? What are the assumptions? Is this based on the actual analysis technique in the study, or just a simple t-test? In the discussion, the authors should clearly point out if the sample size assumptions were actually realized in the clinical trial so we can figure out if the study was over- or under- powered.

2. Lots of great regression models, and not a single word about assumptions of the model, or whether these assumptions were tested or met.

6. PLOS authors have the option to publish the peer review history of their article (what does this mean?). If published, this will include your full peer review and any attached files.

Reviewer #1: No

Reviewer #2: No

---

## [Author Response · Author response to Decision Letter 0]

22 Dec 2020

Dear Editor,

Thank you for considering our paper for publication in PlosOne.

You will find below our responses to each point raised by the academic editor and reviewer(s). 

Best regards

 

Academic Editor’s comments

We have done so

2. Thank you for submitting your clinical trial to PLOS ONE and for providing the name of the registry and the registration number. The information in the registry entry suggests that your trial was registered after patient recruitment began. PLOS ONE strongly encourages authors to register all trials before recruiting the first participant in a study.

1) your reasons for your delay in registering this study (after enrolment of participants started)

All the protocols were originally written in French. The submission to clinicaltrials.gov needed translation of our protocol and the review process went late because of administrative delay in our hospital and unavailability of some actors participating in the consensus of decisions regarding the protocol. We decided not to delay the research, and the manuscript makes it plain that study approval by the ethics committee was in March of 2018 and procedures began in August of 2018. Registration at "clinicaltrials.gov" is not mandated by Tunisian law, which covered all legal aspects of this study. 

2) confirmation that all related trials are registered by stating: “The authors confirm that all ongoing and related trials for this drug/intervention are registered”.

Please also ensure you report the date at which the ethics committee approved the study as well as the complete date range for patient recruitment and follow-up in the Methods section of your manuscript.

We have done so

3. Thank you for including your ethics statement: "ethics committee mongi slim university hospital , la marsa tunisia

approval number 05/2018

written consent"

Please amend your current ethics statement to confirm that your named institutional review board or ethics committee specifically approved this study.

We have done so

"NO authors have competing interests"

We note that one or more of the authors are employed by a commercial company: Ramsay Healthcare France.

The Ramsay Healthcare Group is a private hospital group. It is not a simple commercial company, but a company owning several private hospitals in Europe. The doctors who work in these hospitals are principally not salaried employees but liberal doctors, and they are not necessarily financed by the private hospital for their research work. In this case, none of the doctors received any funding from Ramsay Healthcare.

4.1. Please provide an amended Funding Statement declaring this commercial affiliation, as well as a statement regarding the Role of Funders in your study. If the funding organization did not play a role in the study design, data collection and analysis, decision to publish, or preparation of the manuscript and only provided financial support in the form of authors' salaries and/or research materials, please review your statements relating to the author contributions, and ensure you have specifically and accurately indicated the role(s) that these authors had in your study. You can update author roles in the Author Contributions section of the online submission form.

None of the authors are salaried of Ramsay Healthcare, but doctors affiliated with Ramsay Healthcare hospitals work as liberal doctors and did not receive any funding for this research protocol. 

4.2. Please also provide an updated Competing Interests Statement declaring this commercial affiliation along with any other relevant declarations relating to employment, consultancy, patents, products in development, or marketed products, etc. 

None of the authors are salaried of Ramsay Healthcare, but doctors affiliated with Ramsay Healthcare hospitals work as liberal doctors and did not receive any funding for this research protocol nor have any competing interest. This does not alter our adherence to PLOS ONE policies on sharing data and materials. 

We included both an updated Funding Statement and Competing Interests Statement in our cover letter, thank you for changing the online submission form on our behalf.

Upon review of these statements, we agree that our data has been sufficiently anonymized and provide it to plosone for unrestricted public access

We have done so.

---

## [Decision Letter · Decision Letter 1]

6 Jan 2021

The extraperitoneal French AmbUlatory cesarean section technique leads to improved pain scores and a faster maternal autonomy compared with the intraperitoneal Misgav Ladach technique: A prospective randomized controlled trial

PONE-D-20-27208R1

Dear Dr. Kaouther Dimassi,

We’re pleased to inform you that your manuscript has been judged scientifically suitable for publication and will be formally accepted for publication once it meets all outstanding technical requirements.

Kind regards,

Georg M. Schmölzer

Academic Editor

PLOS ONE

Additional Editor Comments (optional):

Reviewers' comments:

Reviewer's Responses to Questions

**Comments to the Author**

1. If the authors have adequately addressed your comments raised in a previous round of review and you feel that this manuscript is now acceptable for publication, you may indicate that here to bypass the “Comments to the Author” section, enter your conflict of interest statement in the “Confidential to Editor” section, and submit your "Accept" recommendation.

Reviewer #1: All comments have been addressed

2. Is the manuscript technically sound, and do the data support the conclusions?

Reviewer #1: Yes

3. Has the statistical analysis been performed appropriately and rigorously? 

Reviewer #1: Yes

4. Have the authors made all data underlying the findings in their manuscript fully available?

Reviewer #1: Yes

5. Is the manuscript presented in an intelligible fashion and written in standard English?

Reviewer #1: Yes

6. Review Comments to the Author

Reviewer #1: (No Response)

7. PLOS authors have the option to publish the peer review history of their article (what does this mean?). If published, this will include your full peer review and any attached files.

Reviewer #1: No

---

## [Editor Report · Acceptance letter]

11 Jan 2021

PONE-D-20-27208R1 

The extraperitoneal French AmbUlatory cesarean section technique leads to improved pain scores and a faster maternal autonomy compared with the intraperitoneal Misgav Ladach technique: A prospective randomized controlled trial 

Dear Dr. dimassi:

I'm pleased to inform you that your manuscript has been deemed suitable for publication in PLOS ONE. Congratulations! Your manuscript is now with our production department. 

Kind regards, 

on behalf of

Dr. Georg M. Schmölzer 

Academic Editor

PLOS ONE